# Optimization of an Impact-Based Frequency Up-Converted Piezoelectric Vibration Energy Harvester for Wearable Devices

**DOI:** 10.3390/s23031391

**Published:** 2023-01-26

**Authors:** Pietro Aceti, Michele Rosso, Raffaele Ardito, Nicola Pienazza, Alessandro Nastro, Marco Baù, Marco Ferrari, Markku Rouvala, Vittorio Ferrari, Alberto Corigliano

**Affiliations:** 1Department of Civil and Environmental Engineering, Polytechnic of Milan, 20133 Milano, Italy; 2Department of Aerospace Science and Technology, Polytechnic of Milan, 20156 Milano, Italy; 3Department of Information Engineering, University of Brescia, 251121 Brescia, Italy; 4Huawei Technologies Oy, FI-0620 Helsinki, Finland

**Keywords:** piezoelectric materials, energy harvesting, mechanical vibrations, impacts, microsystems

## Abstract

This work presents a novel development of the impact-based mechanism for piezoelectric vibration energy harvesters. More precisely, the effect of an impacting mass on a cantilever piezoelectric transducer is studied both in terms of the tip mass value attached to the cantilever and impact position to find an optimal condition for power extraction. At first, the study is carried out by means of parametric analyses at varying tip mass and impact position on a unimorph MEMS cantilever, and a suitable physical interpretation of the associated electromechanical response is given. The effect of multiple impacts is also considered. From the analysis, it emerges that the most effective configuration, in terms of power output, is an impact at the cantilever tip without a tip mass. By changing the value of the tip mass, a sub-optimal impact position along the beam axis can also be identified. Moreover, the effect of a tip mass is deleterious on the power performance, contrary to the well-known case of a resonant energy harvester. A mesoscale prototype with a bimorph transducer is fabricated and tested to validate the computational models. The comparison shows a good agreement between numerical models and the experiments. The proposed approach is promising in the field of consumer electronics, such as wearable devices, in which the impact-based device moves at the frequencies of human movement and is much lower than those of microsystems.

## 1. Introduction

Nowadays, sensors and actuators in the form of micro-electro-mechanical systems (MEMS) are ubiquitous, not only in personal devices, such as smartwatches, smartphones, and fitness trackers but also in the development of smart systems (internet of things, sensor networks) [1]. In all cases, the attention is focused on low-power sensors to reduce battery consumption. In this perspective, the adoption of an external source of energy can be highly beneficial to further increase the battery life and, in the limited case, for the development of fully autonomous sensors. One of the most attractive sources of energy is human motion: we develop kinetic energy, dissipating more than 100 W [2], through our movements [3], not only while walking and running [4,5] but even while gesturing. A large amount of wearable biofeedback devices have been developed in recent decades [6] and also recently [7], a useful solution to improve the power autonomy of sensors and small devices could be represented by energy harvesters (EHs) that are able to convert the mechanical energy into an electrical one [8,9,10]. From the literature, it is clear that piezoelectric transducers, typically in the form of beams or plates, are effective when they oscillate at the resonant frequency [11,12,13,14,15]. Human movement is characterised by a broad frequency spectrum dominated by low-frequency components in the range of 2–5 Hz [16], whereas the resonance frequency of a micro cantilever beam is around hundreds or thousands of Hertz. This frequency mismatch leads to the suboptimal effectiveness of the harvester. To overcome this issue, different studies in the literature propose the adoption of frequency up-conversion [17], which can be achieved via the non-linear behaviour of structural elements [12,13,14,15,16,17,18,19,20], buckling [21,22,23,24], and magnetic interaction [25,26,27,28]. In other cases, with the purpose of increasing the scavenged energy, arrays of resonant converters are proposed [29,30,31,32], and the adoption of special metamaterials is introduced [33,34,35]. On the other hand, for ultra-low ambient frequency, dry reciprocating friction is exploited [36]. The aim of frequency up-conversion is to change the frequency content of the input vibration, bridging the gap with respect to the first natural frequency of the piezoelectric converter. Other solutions aim to shift the natural frequency of the structure by changing the stiffness of the transducer, introducing some substructures [37], or changing the geometry [38]. Although these methods improve the electro-mechanical transduction effectiveness of piezoelectric transducers, they do not exploit in any way the random component of human movements, which is much more accentuated than the periodical one.

In impact-based EH, piezoelectric beams can convert into electrical energy not only the energy related to a periodic excitation but also the random excitation of human movements, such as quick wrist movements during a walk. In the literature, some macroscopic examples that involve these impacts are given in [39,40,41,42,43]. In [39], an analytical description of a device with two macroscopic piezoelectric beams and a seismic mass is presented. However, the size of the device and the required oscillation path (10 cm) are not suitable for wearable applications. This paper investigates, through numerical models and experimental results, the impact effect on devices containing a single piezoelectric cantilever transducer with dimensions suitable for wearable applications.

In the literature, various papers such as [40] and [41] show a configuration in which the cantilever beam presents a proof mass at its free edge. In this work, a parametric study was performed to find the best impact position and dimension of the proof mass and also consider the effect of multiple impacts on a micro piezoelectric beam.

The paper is organized as follows.

In Section 2, the operation principle is described together with a numerical model developed using the finite element code ABAQUS-Simulia^®^, coupled with a proper Fortran subroutine that simulates the external electrical resistive circuit.

An optimization process is carried out in Section 3 to improve the output power, changing the position of the impact and adding different tip masses at the free edge of the transducer. The best result in terms of instantaneous and mean converted power is obtained by a piezoelectric beam without tip mass, impacted at the free edge.

Section 4 contains a comparison between the simulation results and experimental tests, which shows the good capability of the numerical tool to reconstruct the physical phenomenon.

Final remarks and conclusions are proposed in Section 5.

## 2. Description of the Operational Principle and Numerical Model

The object of the study is referred to in a finite volume of 22 mm × 6 mm × 1.8 mm, which is suitable for a wearable device. The considered volume contains a piezoelectric beam, representing the energy transducer for the conversion of mechanical energy into an electrical one. The considered scenario exploits the free fall of an impact mass on the free edge of a piezoelectric beam. The working principle, depicted in Figure 1, concerns the power production of the considered piezoelectric cantilever beam impacted by a steel sphere with a mass of 110 mg, which is free to move along a rectilinear path under a uniform acceleration of 1 g. As shown in Figure 1, the sphere movement is activated by the action of gravity after a rotation of 180° around the y-axis. This simple device exploits the impact mechanism of the free steel ball on the transducer. The analysis allows the possibility to have multiple impacts on the cantilever. For the sake of simplicity, the rotation must be thought to be instantaneous. In the considered conditions, the impact velocity is equal to 0.533 m/s and computed considering the steel sphere travelling a distance of 14.5 mm inside the available volume.

The unimorph piezoelectric cantilever transducer used in the analyses is shown in Figure 2. It is composed of a thin film of piezoceramic material PZT (Lead Zirconate Titanate) on a silicon substrate. The material and geometrical features suited for the realization at the MEMS scale are summarized in Table 1 and Table 2, respectively.

In Table 1, the piezoelectric and dielectric constants are given in accordance with the IEEE standards for piezoelectricity [44], which adopts the Voigt notation. The constitutive law is written in the *e-form*, with the hypothesis of neglecting the coupling between the electrical field and the shear stress. The polarization vector is aligned with the *z* axis, so that the piezoelectric coupling matrix is written as follows:(1)$$e=\left[\begin{array}{cccccc} 0 & 0 & 0 & 0 & 0 & 0\\ 0 & 0 & 0 & 0 & 0 & 0\\ e_{31} & e_{32} & e_{33} & 0 & 0 & 0 \end{array}\right]$$

Note that in the matrix (1), *e*_31_ = *e*_32_ because of the symmetry of the material.

The finite element model contains three-dimensional finite elements both for silicon and the piezoelectric material. Fully coupled dynamic analyses were carried out. It is important to point out that mechanical damping is introduced to represent a typical value of the mechanical quality factor for the entire device *Q_M_ =* 500 [1]. This value can be reached in a standard manufacturing process for MEMS, which requires a controlled environment. As a matter of fact, the mechanical quality factor influences the electromechanical transduction ability [45]; thus, different values could change the output power but not the overall qualitative behaviour. It is possible to obtain the damping ratio $\zeta_{M}\;$for a lumped parameter model starting from the quality factor as:(2)$$\zeta_{M}=\frac{1}{2Q_{M}}$$

Material damping is introduced in the finite element model through Rayleigh’s formula:(3)$$C=\alpha_{R}M+\beta_{R}K\;$$

In Equation (3) ***C****, **M**,* and ***K*** are the damping, mass and stiffness matrices of the structure, respectively. $\alpha_{R}$ and $\beta_{R}$ are the Rayleigh damping coefficients. By means of simple algebraic manipulations, the vibrational mode *n,* can be obtained as:(4)$$2\zeta_{M,n}=\frac{\alpha_{R}}{\omega_{n}}+\beta_{R}\omega_{n}$$

In Equation (4), $\omega_{n}$ is the natural angular frequency of the mode *n*. In the Rayleigh damping model, it is known that the coefficient $\alpha_{R}$ concerns the dissipation of low-frequency vibration modes. On the contrary, $\beta_{R}$ relates to the damping for high-frequency vibrations modes. Assuming that the dynamic response of the structure is mainly due to the first mode, the Rayleigh damping coefficients can be chosen as:(5)$$\beta_{R}=0,\;\;\alpha_{R}=2\zeta_{M}\omega_{r}=\frac{\omega_{r}}{Q_{M}}\;$$

In Equation (5) $\omega_{r}$ is the natural angular frequency of the first mode, which has a value of 5504 rad/s, which is computed through the finite element modal analysis. Accounting for the assumed quality factor *Q_M_ =* 500, the value $\alpha_{R}=$ 11.00 rad/s is obtained.

The piezoelectric element is connected to a simple electric circuit with a resistive load that is characterised by the optimal resistance of 4.67 kΩ. This value is the optimal load for energy scavenging [12] and can be obtained through the formula:(6)$$R_{opt}=\frac{1}{C_{0}\omega_{r}}\;$$

In Equation (6), *C*_0_ is the capacitance of the piezoelectric layer computed as:(7)$$C_{0}=\frac{S\varepsilon_{0}\varepsilon_{r}}{d}\;$$
where *d* is the piezoelectric thickness, and *S* is the piezoelectric surface in the plane orthogonal to the *z-*axis. Finally, $\varepsilon_{0}=$ 8.854 × 10^−12^ F/m and $\varepsilon_{r}=$2000 are the vacuum and relative permittivity of the piezoelectric material, respectively. It is important to highlight that in the computational model, for the sake of simplicity, the dielectric losses (the loss tangent factor) are neglected.

The graph in Figure 3 shows the electric instantaneous power vs. time plot, namely the impact of a steel sphere, as shown in Figure 1. The time is set to zero at the moment of the impact of the sphere on the beam tip. In the subsequent time instants, the kinetic energy of the sphere is transformed into the strain energy of the beam, which attains the maximum deformation at about 4.5 ms. In this time interval, tiny energy production is observed, with a maximum of about 2 ms. The sphere leaves the beam at about 7 ms, and the free oscillation of the cantilever begins thereof, with the typical decay in agreement with the chosen quality factor. The peak power estimated as *V*^2^/*R_opt_* was about 120 μW.

The sphere spends 54 ms to cover its rectilinear path. Taking into account the inverse path, the subsequent impact may occur after 108 ms. The analysis reported in Figure 3 confirms that the free vibration is completely damped out at that time. Thus, integrating the instantaneous power, the energy on the time interval between two subsequent impacts is 0.550 μJ. Almost 10 impacts may happen in one second so that the cumulated energy might reach about 5 μJ.

From the presented numerical results, the use of impacts that exploit the random input given by human motion appears promising in the piezoelectric vibration energy harvesting field. It is also important to put in evidence that the considered transducer has no tip mass; this particular point is the object of the discussion presented in the subsequent Section 3.

## 3. Optimization of a Cantilever Beam at the Microscale

In this section, an attempt is made to improve the performance of the microscale cantilever beam considered in Section 2 by changing its mechanical properties. From the literature, it is clear that adding a tip mass on the piezoelectric cantilever beam oscillating at its resonance frequency improves the output power at the parity of input acceleration [2].

A series of numerical simulations with different tip masses was performed in order to investigate the variation in response to the impact-based case. Five different amounts of mass were added to the tip as a fraction of the mass *M* of the impacting steel body. In particular, the values *M, M*/2*, M*/3*, M*/4*,* and *M*/10 are considered; the corresponding cases are, respectively, named TM1, TM2, TM3, TM4, and TM10. Moreover, the case without a tip mass is renamed WTM. During the investigation, the impact position along the length *L* of the cantilever beam, which affects the response, was considered. For each tip mass, three impact positions were analysed: *L* (3000 μm), ¾*L* (2250 μm), and *L/*2 (1500 μm) from the clamped section. The tip mass, for each case, is designed with a square cross-section, width equal to the beam width (*w* = 1500 μm), and employs the same material as the impacting body. The tip mass is clamped just beneath the structural layer, as shown in Figure 4 (related to the case TM10).

It is important to note that the variation in the tip mass causes a variation in the eigen-frequency of the electromechanical system; therefore, according to Equations (5) and (6), the damping coefficient $\alpha_{R}$ and the optimal electric resistance *R_opt_* change for each case. These parameters are summarized in Table 3 for each model, together with the corresponding size of the cross section.

The parametric study shows that the impact localized in different positions with respect to the end of the beam and activated higher vibration modes. However, the activation of these modes is almost never associated with a higher power extraction. This highlights the advantage of causing the impact at the tip of the beam. In fact, when the impact occurs near the free edge, the behaviour of the cantilever is similar to a free vibration due to an imposed initial displacement at the free edge: this is governed by the low-frequency modes of the beam. Conversely, when the impact occurs close to the constrained edge, the effect of the impact on the cantilever is more similar to an impulse creating a multimodal vibration. In fact, for a cantilever beam, the driving point residue [46] of the points near the constraint suggests the more suitable ability of these points to transfer energy and frequency content. The multi-modal excitation may appear beneficial, but the impact location influences the possibility of multiple impacts on the vibrating beam with the sphere [3]: the double impact happens both for the case with an impact position at ¾*L* and for the case with an impact position at *L*/2. These aspects emerge from the numerical simulations and are here presented only for the TM10 for the sake of conciseness (Figure 5, Figure 6 and Figure 7). The response for the impact at the tip is dominated by the first oscillation mode, whereas in the two other cases, the effect of higher frequency modes is clearly visible, but the presence of multiple impacts limits the free oscillation and, as a consequence, the power production.

Having examined the effect of the impact position, the role of the size of the tip mass for the impact-based energy harvester is now presented. Figure 8 shows the responses in terms of the mean power (computed over a time duration of 30 ms) of all the cases. In the case of impact at the free edge, it can be noted that the power output increased at the decreasing tip mass, contrary to what occurs in the case of a linear resonant device. If the beam is forced with a harmonic input at a given acceleration, the presence of the mass is beneficial since it adds an inertial contribution that amplifies the oscillations. The same interpretation can be reached from a modal-dynamics point of view: by exciting the first mode of the structure, the mass attached to the tip increases the participating mass to the motion; therefore, energy is involved. This is not true in the case of impacts: the added mass creates resistance to the impacting body by introducing an inertial contribution due to the conservation of momentum. This causes a decrease in deformation and, so, involves electric power.

The impact at ¾ L always shows a lower power generation compared to the other cases due to the fact that the impact point is very close to the node of the second bending mode of vibration and, most importantly, since multiple impacts happen before the sphere detaches the beam. The multiple impacts induce a strong limitation of the beam movement so that the power production is small.

The impact at the midpoint of the beam length shows a non-monotonic power generation again in view of the presence of multiple impacts: as far as the tip mass is low (or absent), the beam rebounds promptly, and the sphere is hit again. For intermediate tip masses (i.e., TM2-TM3-TM4-TM10), the second impact does not happen, and the power generation is satisfactory through a combination of the first and the second bending modes. Finally, for big masses, the same trend as for the tip impact is recovered.

In general, the promising response of the impact-based energy harvester is also interesting because the energy flow mechanism is simple: during the impact, the beam is deformed, transforming the kinetic energy of the sphere into deformation energy. As soon as the impacting body is released, the beam enters a free vibration regime at its eigen-frequency and amplitude proportional to the deformation of the beam at the instant of detachment with the impacting body. The interaction between the sphere and cantilever after the first impact is deleterious since it decreases the amplitude of the free oscillation. The parametric study shows that the best situation for a cantilever micro piezoelectric beam occurs for a beam without proof mass and an impact on the free edge. The output power is shown in Figure 3.

## 4. Numerical Model Validation on a Macroscale Prototype

To validate the numerical model discussed in Section 2 and Section 3, a macroscale prototype has been fabricated and tested. The experiment and its comparison with the predicted results also preserve their importance at the micro-scale. The transducer used in the experiment is a commercial bimorph cantilever (Figure 9, RS 285-784, RS Component^®^, Corby, UK), and its physical and geometrical data are summarized in Table 4 and Table 5, respectively. In this case, the piezoelectric beam has a core made of titanium. The piezoelectric layers are connected in series and have opposite polarities.

The prototype is composed of a mechanical frame, a steel sphere, and the mentioned piezoelectric beam. The sphere has a diameter of 7 mm and a weight of 1.4 g.

The entire mechanical frame of the prototype was first designed with SolidWorks^©^ software and then a built-in laboratory. The construction operations were carried out starting from 1.7 mm thick FR4 plates, which were then cut with a numerical control pantograph (CNC procedure) according to the design. The frame is basically composed of three parts: the base, collar, and anchors, as illustrated in Figure 10. The components of the frame are assembled in the order (a), (b), and (c) of Figure 10 from the bottom to the top. The anchors create a proper zone to guarantee the clamp boundary condition for the cantilever.

The clamp is created by gluing the beam to the anchors with a cyanoacrylate adhesive. To realize an efficient clamp and to create the electrical connections, a 6 mm portion of the beam length is inserted in the anchor.

The remaining part of the beam, exploited as the impact zone, is about 9 mm. The experiment consists of making the impact of the steel sphere on the piezoelectric transducer’s tip after a stroke of 58.5 mm, which is obtained by tilting the prototype with an angle θ = 6°. To ensure the impact on the cantilever tip, a longitudinal track is present on the case bottom plate to guarantee a mechanical guide for the motion of the ball, as reported in Figure 11, showing the whole experimental set-up.

The acquisition of the output voltage generated by the cantilever is made with the Keysight MSOX3014A oscilloscope. Voltage measurements are taken across an optimal resistance load of *R_opt_ =* 85 kΩ in parallel with a resistance of *R_osc_ =* 1 MΩ introduced by the oscilloscope. The value of *C_p_* is equal to 770 pF, as computed through Equation (7) by summing the contributions of two piezoelectric layers. In Figure 12, the electrical model of the piezoelectric beam is presented.

According to the geometry of the prototype, a finite element model is created. To model the clamp in the numerical simulation, the copper anchors are placed in their position as two fully constrained rigid bodies of size 2 mm × 2 mm × 1.5 mm. The surface portions of the beam in contact with the blocks are constrained with them through a tie interaction (Figure 13). The glue used in the prototype during the construction phase is neglected. This approximation introduces some difference during the impact but leaves unchanged the overall behaviour during the free oscillation.

The impact steel ball of radius *r* = 7 mm is modelled as a simple rigid body with a mass *M* = 1.4 g and with rotational inertia 2/5$\;Mr^{2}$ = 6860${\;\mathrm{kg}\;\mu m}^{2}$. To save time and computing power, not all the free fall of the impacting body is modelled, but an initial speed of 0.35 m/s is imposed on the body in a position just above the tip beam according to real motion. The initial speed is computed on the basis of the path followed by the sphere.

The electric load resistance *R*, inserted in the Fortran subroutine, is the parallel between the optimal electrical load and oscilloscope electrical load. It is obtained as follows:(8)$$R=\frac{R_{opt}R_{osc}}{R_{opt}+R_{osc}}=78.34\;k\Omega \;$$

Comparing the numerical and experimental results, it is possible to deduce that the numerical model gives a good estimation of the coupled electromechanical behaviour of the beam during an impact. Figure 14 shows the numerical and experimental voltages as a function of time.

There was a difference in the time response of the first peak corresponding to the deformations that occurred due to the impact. This effect is probably due to the fact that the glue was not able to realize an ideal clamp, and this is not introduced in the numerical simulations. The free motion after the impact is well reconstructed in terms of frequency and amplitude. The decay is slower in the experiment: this is connected to the overestimation of damping in the numerical model.

From the voltage *V*_R_, which is the voltage across the load resistor for the considered situation of impact and free vibration, it is possible to derive instantaneous power:(9)$$P=\frac{V_{R}^{2}}{R}$$

Figure 15 shows the comparison between the experimental (red line) and the numerical (blue line) results in terms of the output power. Though a mismatch is visible in the impact phase, for the same reasons commented above, there is a satisfactory agreement in the free vibration. In conclusion, it is possible to state that the numerical models capture the physical phenomenon of interest with a good approximation.

## 5. Conclusions

A numerical-experimental study of impact-based energy harvesting systems is presented for a piezoelectric cantilever.

A device that can be integrated into a smartwatch or a smartphone was first studied. The designed device occupies a volume with dimensions 22 mm × 6 mm × 1.8 mm and is able to harvest a mean power of 26.39 μW over a time interval of 30 ms using a single piezoelectric converter.

A parametric study at a varying entity of the tip mass of the transducer and impact position was then carried out, and a physical interpretation of the numerical outcomes was proposed. It emerged that it is advantageous that the impact occurs at the free end of the transducer so that the effect of the impact on the transducer behaviour is similar to an imposed initial displacement applied on the free edge of the beam. The approach of the impact-based harvester is useful and promising in the framework of low-frequency applications and for random signals, as in the case of consumer devices that are subject to human motion (smartphones, smartwatches, etc.). It is worth noting that human motion is typically characterized by random movements (e.g., while gesturing) or by a sequence of pulses at low frequency (during running). In both cases, it would be rather difficult to steadily excite a resonant device, whereas the impact-based device can work properly.

To validate the results obtained with the numerical model, a simple macroscale prototype was fabricated and tested. The comparison between the numerical models and experiments shows a good agreement.

It would be interesting in the future to study reliability issues [47]. Piezoceramics are typically brittle: the repeated impacts could induce ruptures. Such a fact could be the main disadvantage of an impact-based energy harvester. Another interesting aspect could be to implement the concept in a real wearable device and test it for human motion activities, as conducted in [29]. It would also be interesting to extend the study to the case of indirect impacts, e.g., the impacts on the substrate on which the piezoelectric transducer is installed. In that way, the impact does not happen on the piezoelectric surface directly, even though the energy transfer could be less effective in view of the elastic wave diffusion in the substrate.

## Figures and Tables

**Figure 1 sensors-23-01391-f001:**
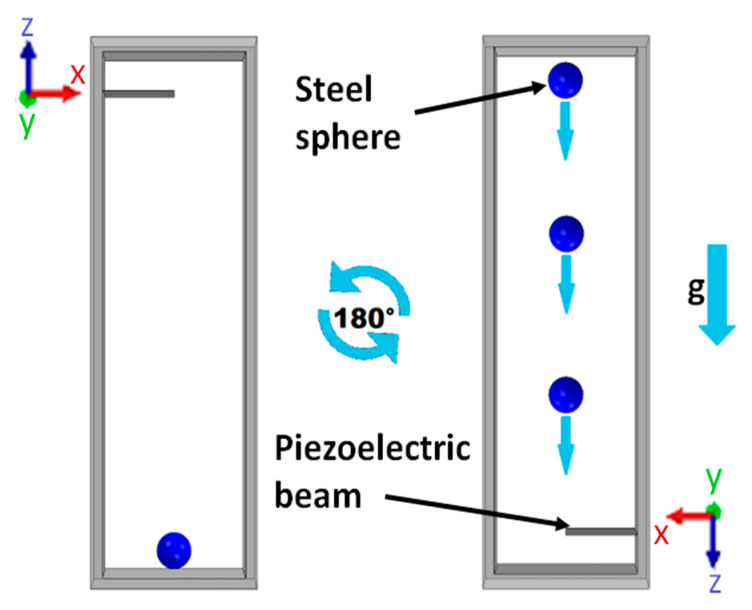
Illustration of the theoretical instantaneous 180° rotation of the device and the impact mechanism.

**Figure 2 sensors-23-01391-f002:**
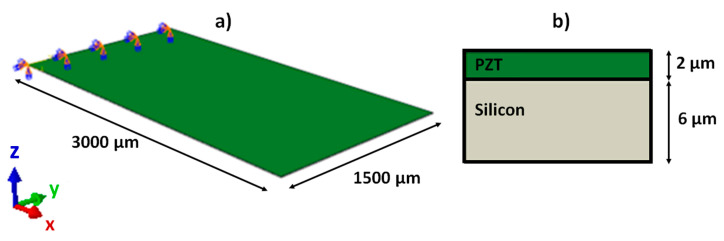
Illustration of the computational model of the piezoelectric transducer. (**a**) 3D view (symbols on the top-left highlight the clamped-in edge), (**b**) Layered cross-section.

**Figure 3 sensors-23-01391-f003:**
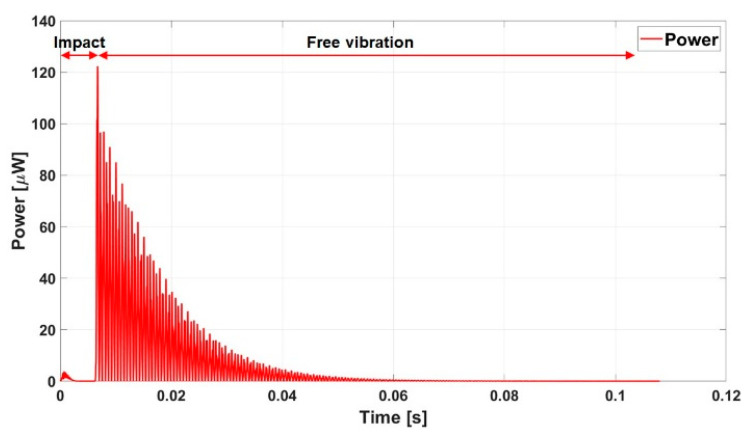
Instantaneous power for an impacted sphere on a piezoelectric beam.

**Figure 4 sensors-23-01391-f004:**
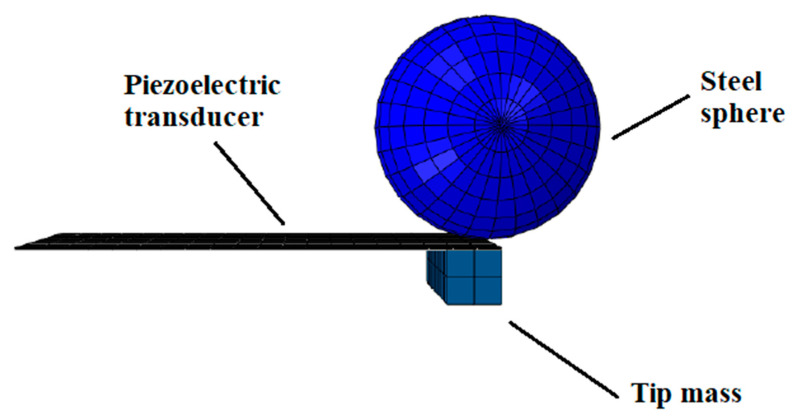
Finite element model at the instant of impact: case with tip mass equal to 1/10 of the sphere mass.

**Figure 5 sensors-23-01391-f005:**
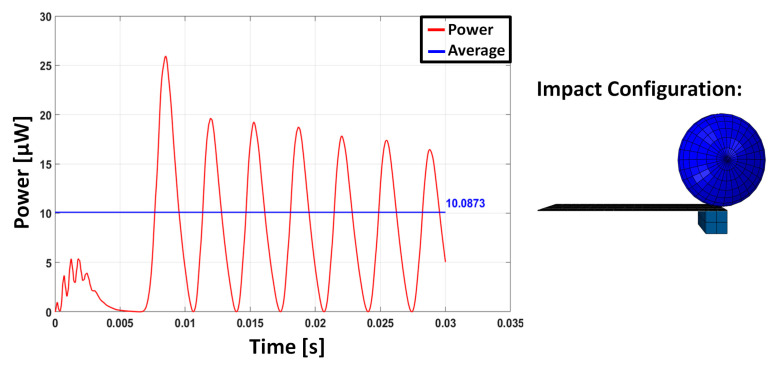
Instantaneous power and average value computed over a time duration of 30 ms in the case of TM10 with impact position L.

**Figure 6 sensors-23-01391-f006:**
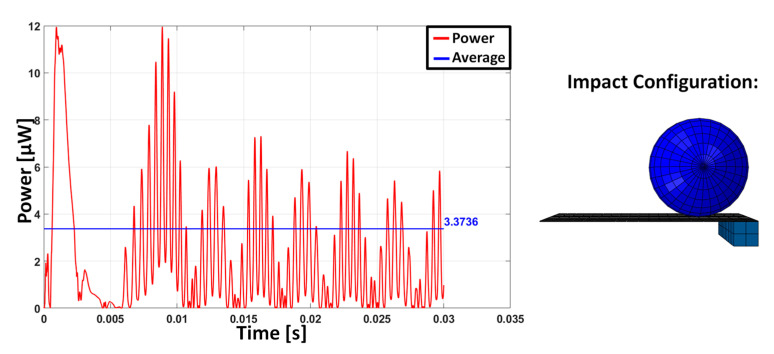
Instantaneous power and average value computed over a time duration of 30 ms in the case of TM10 with impact position ¾ L.

**Figure 7 sensors-23-01391-f007:**
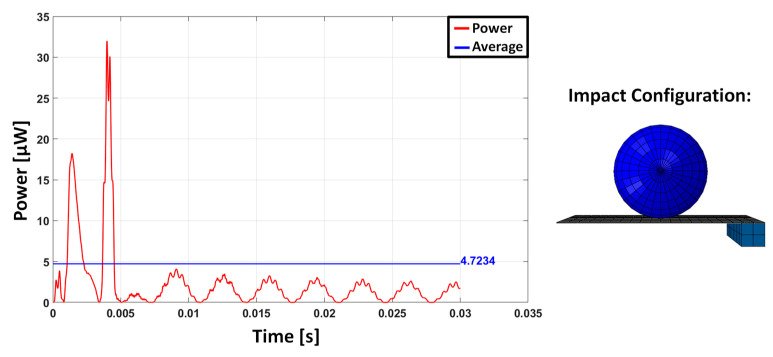
Instantaneous power, and average value computed over a time duration of 30 ms in the case of TM10 with impact position L/2.

**Figure 8 sensors-23-01391-f008:**
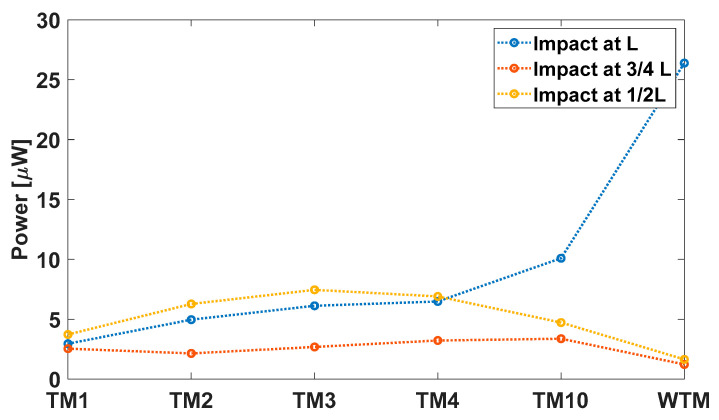
Average power plot of the parametric analyses for different values of tip mass placed on the transducer.

**Figure 9 sensors-23-01391-f009:**
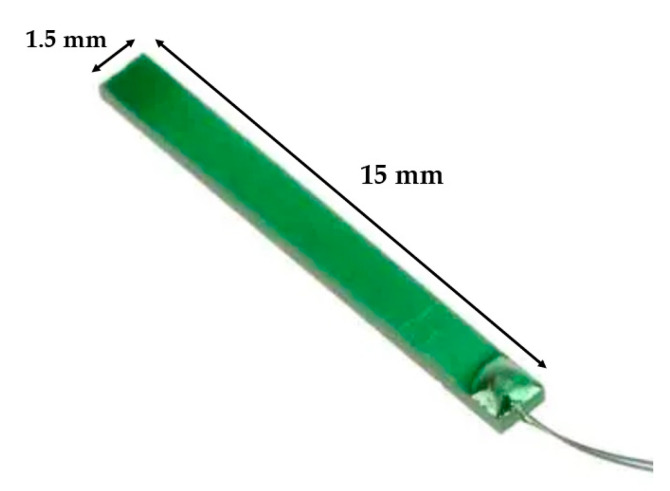
Bimorph piezoelectric beam used in the experimental setup: RS 285-784, RS Component. Dimensions: length 15 mm, width 1.5 mm, and thickness 0.6 mm.

**Figure 10 sensors-23-01391-f010:**
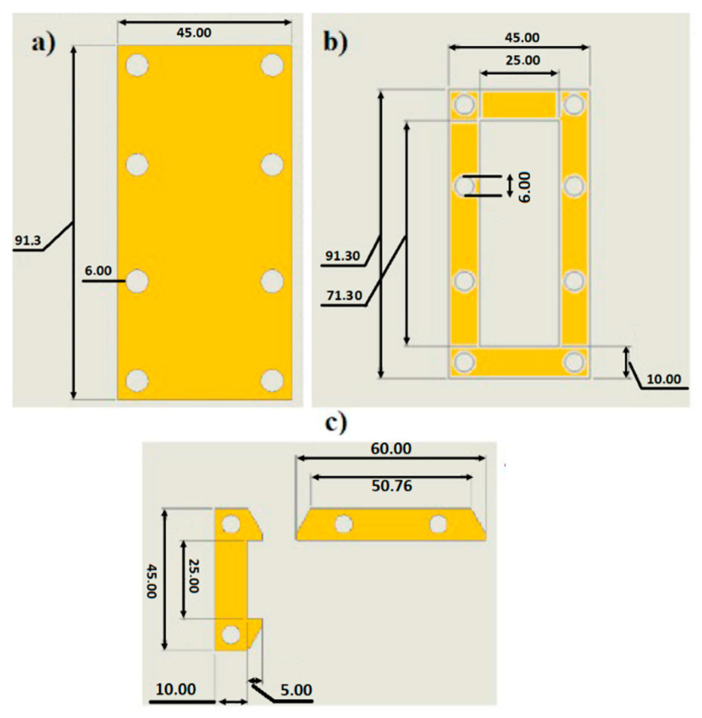
Components of the mechanical frame: (**a**) Base (**b**) Collar (**c**) Anchors. Dimensions are in mm.

**Figure 11 sensors-23-01391-f011:**
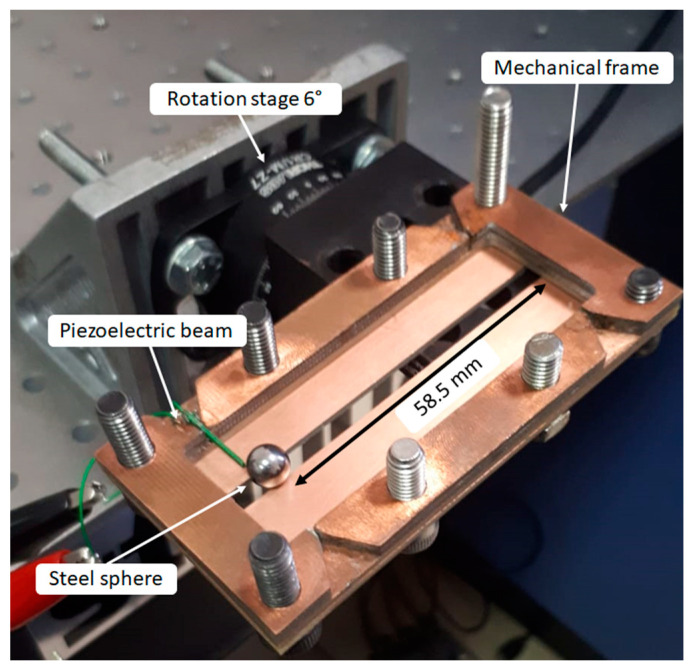
Experimental setup: mechanical frame installed on the rotation stage.

**Figure 12 sensors-23-01391-f012:**
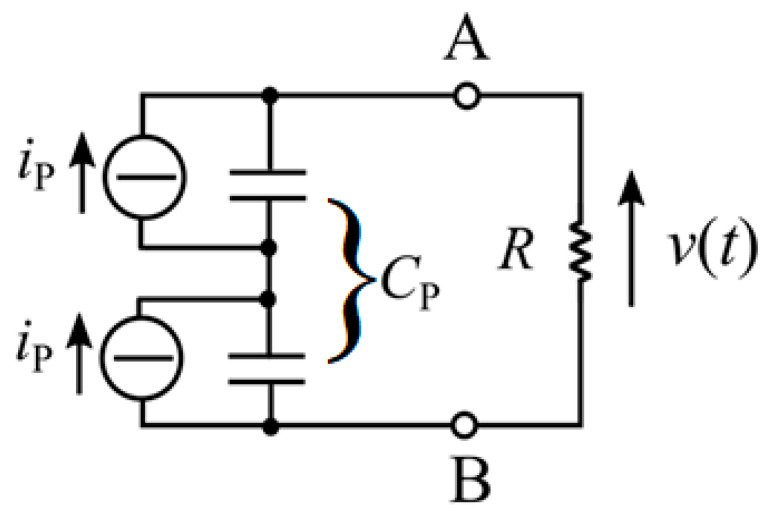
Electrical model of the piezoelectric energy harvester.

**Figure 13 sensors-23-01391-f013:**
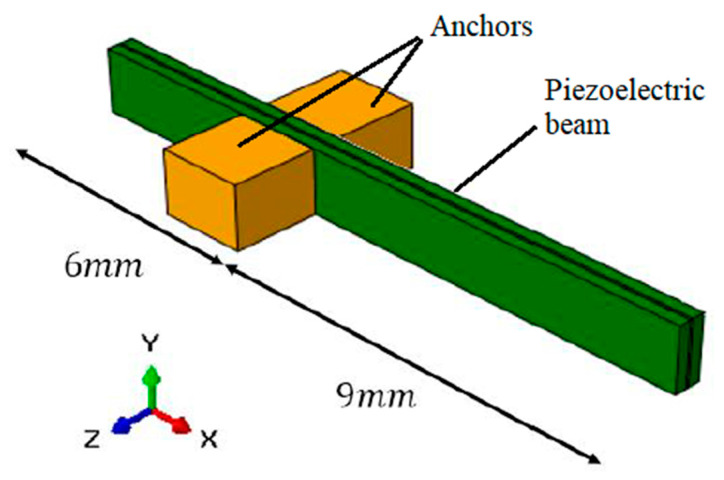
Illustration of the piezoelectric transducer and the anchors.

**Figure 14 sensors-23-01391-f014:**
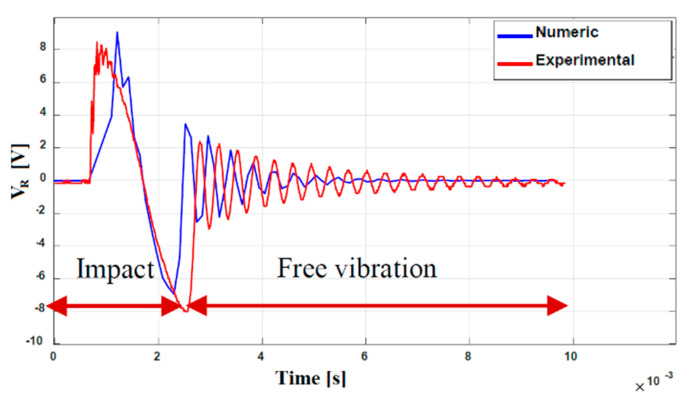
Experimental and numerical voltage vs. time plots.

**Figure 15 sensors-23-01391-f015:**
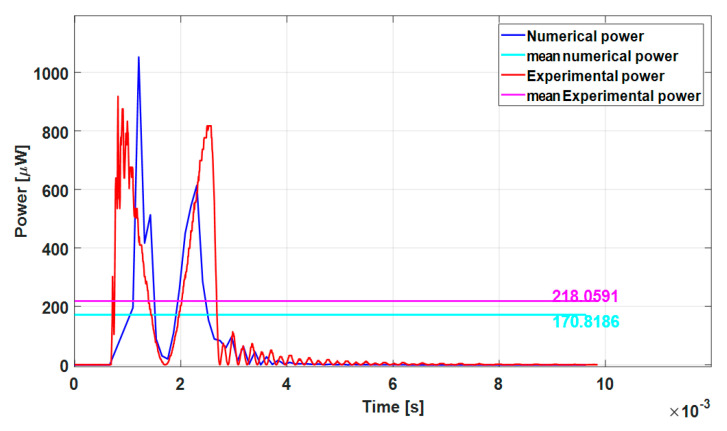
Experimental and numerical power vs. time plots.

**Table 1 sensors-23-01391-t001:** Physical properties of the considered materials,$\;\varepsilon_{0}=$ 8.854 × 10^−12^ F/m.

Material	Mass Density [kg/m^3^]	Young’s Modulus [GPa]	Poisson’s Ratio $\left[-\right]$	*e*_31_ * [N/(Vm)]	*e*_33_ * [N/(Vm)]	$\mathrm{Relative}\;\mathrm{Dielectric}\;\mathrm{Constant}\;(\mathrm{Static})\;\varepsilon_{r}\;\left[-\right]$
Silicon	2330	148	0.3	-	-	-
PZT, thin film	7700	100	0.3	−12	20	2000

* Piezoelectric constants “e-form”.

**Table 2 sensors-23-01391-t002:** Geometrical features of the cantilever beam.

Material	In-plane Dimensions [μm × μm]	Thickness [μm]
Silicon	3000 × 1500	6
PZT, thin film	3000 × 1500	2

**Table 3 sensors-23-01391-t003:** Cases considered in the parametric study.

CASE	Cross-section Size of the Tip Mass [μm × μm]	Eigen-Frequency f_r_ [Hz]	Rayleigh Coefficient$\;\alpha_{R}[-]$	Optimal Electrical Resistance *R_opt_* [kΩ]
TM1	1070 × 1070	52.59	0.66	76.03
TM2	750 × 750	70.68	0.88	56.57
TM3	618 × 618	83.43	1.05	47.92
TM4	535 × 535	94.62	1.19	42.26
TM10	338 × 338	142.26	1.79	28.11
WTM	-	875.97	10.83	4.64

**Table 4 sensors-23-01391-t004:** Physical properties of the involved materials ε_0_ = 8.854 × 10^−12^ F/m.

Material	Mass Density [kg/m^3^]	Young’s Modulus [GPa]	Poisson’s Ratio [-]	d_31_ * $\left[\frac{pC}{N}\right]$	Relative Dielectric Constant (Static) $\varepsilon_{r}\;\left[-\right]$
Titanium	4500	115	0.3	-	-
NTK code MT-11	7500	60	0.3	215	2000

***** Piezoelectric coefficient “d-form”.

**Table 5 sensors-23-01391-t005:** Geometrical features of the cantilever beam.

Material	In-plane Dimensions [mm × mm]	Thickness $\left[\mu m\right]$
Titanium	1.5 × 15	65
NTK code MT-11	1.5 × 15	280 (for each layer)

## Data Availability

The data presented in this study are available on request from the corresponding author.

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
