# Peer review of "Optimization of an Impact-Based Frequency Up-Converted Piezoelectric Vibration Energy Harvester for Wearable Devices"

_sensors, 2023, doi:10.3390/s23031391_

Round 1

Reviewer 1 Report

Authors analyzed impact-based operation mechanism for piezoelectric vibrator and then, provide some optimization techniques using several piezoelevtric parameters. English grammar looks fine. The manuscript is well written. Theoretical and simulated results looks good. Therefore, the manuscript can be minor revision if authors answered the questions as below.

1. Please provide city and country information for conference papers.

2. Please use abbreviated journal names in reference section.

3. Figures 1 and 2 qualiy looks bad to be seen.

4. Tables 1,2,4,5 format are wrong according to MDPI format.

5. Figures 5-7 looks unclear because I guess it is caused by the unclear fonts.

6. What are the length and width sizes in Figure 9 ?

7. Authors had better emphasize the novely in Abstract section.

8. Please provide ref. for the sentence (As a matter of fact, the mechanical quality factor influences the~) with ref. (https://www.sciencedirect.com/science/article/abs/pii/S0263224116306157).

8. In Line 22, consumer electronics wearable devices => consumer electronics like wearable devices.

9. Please change equation to Equation in Line 137.

10. In Figures 5 and 6, there is "Average". How to obtain "Average" ? It looks like not in the middle point.

11. I am wondering how to obtain the numerical graph in Figure 4 and which Equations authors used.

12. Data availibity section is missing.

13. Why the dynamic response of the structure is mainly due to the first mode ?

14. If possible, please describe how to obtain 78.34 k ohm in Equation (8).

15. Authors mentioned that in Line 332, Vr is the voltage. Is that voltage at resonance ?

Author Response

Dear reviewer,

We wish to thank you for the time dedicated to the careful revision of our paper. Attached you can find our answers and comments.

Best Regards
Pietro Aceti

Reviewer 2 Report

.

Author Response

(The authors gave the same response as above.)

Round 2

Reviewer 2 Report

.